



# Compound high temperature and low chlorophyll extremes in the ocean over the satellite period

Natacha Le Grix[1,2], Jakob Zscheischler[1,2,3], Charlotte Laufkötter[1,2], Cecile S. Rousseaux[4,5], and Thomas L. Frölicher[1,2]

[1]Climate and Environmental Physics, Physics Institute, University of Bern, Bern, Switzerland.
[2]Oeschger Centre for Climate Change Research, University of Bern, Bern, Switzerland.
[3]Department for Computational Hydrosystems, Centre for Environmental Research – UFZ, Leipzig, Germany.
[4]Global Modeling and Assimilation Office, NASA Goddard Space Flight Center, Greenbelt, USA.
[5]Universities Space Research Association, Columbia, USA.

**Correspondence:** Natacha Le Grix (natacha.legrix@climate.unibe.ch)

**Abstract.** Extreme events severely impact marine organisms and ecosystems. Of particular concern are compound events, i.e., when conditions are extreme for multiple potential ecosystem stressors such as temperature and chlorophyll. Yet, little is known about the occurrence, intensity and duration of such compound high temperature (aka marine heatwaves - MHWs) and low chlorophyll (LChl) extreme events, whether their distributions have changed in the past decades and what the potential drivers

are. Here we use satellite-based sea surface temperature and chlorophyll concentration estimates to provide a first assessment of such compound extreme events. We reveal hotspots of compound MHW and LChl events in the equatorial Pacific, along the boundaries of the subtropical gyres, in the northern Indian Ocean, and around Antarctica. In these regions, compound events that typically last one week occur three to seven times more often than expected under the assumption of independence between MHWs and LChl events. The occurrence of compound MHW and LChl events varies on seasonal to interannual timescales.

At the seasonal timescale, most compound events occur in summer in both hemispheres. At the interannual time-scale, the frequency of compound MHW and LChl events is strongly modulated by large-scale modes of natural climate variability such as the El Niño-Southern Oscillation, whose positive phase is associated with increased compound event occurrence in the eastern equatorial Pacific and in the Indian Ocean by a factor of up to four. Our results provide a first understanding of where, when and why compound MHW and LChl events occur. Further studies are needed to identify the exact physical and biological

drivers of these potentially harmful events in the ocean and their evolution under global warming.

## 1   Introduction

Over the last few decades, ocean extreme events, such as marine heatwaves (MHWs), have occurred in all ocean basins (Fig. 1a) (Frölicher and Laufkötter, 2018; Hobday et al., 2018; Laufkötter et al., 2020). Alongside the long-term warming of the global ocean (Cheng et al., 2017), the number of MHW days has doubled between 1982 and 2016 (Frölicher et al., 2018; Oliver

et al., 2018) and is projected to increase strongly under continued global warming (Frölicher et al., 2018). MHWs have already negatively impacted many key habitats (Smale et al., 2019), including seagrass meadows (Marba and Duarte, 2010; Thomson





et al., 2015), kelp forests (Wernberg et al., 2013; Smale et al., 2019) and coral reefs (Hughes et al., 2018). Changes in extreme conditions are also expected in the concentration of phytoplankton, that regulate key biogeochemical processes such as ocean carbon uptake and export, and form the bases of the aquatic food web, but so far less is known about extremes in the abundance
of these species.

An emerging concern are compound events, i.e. situations where more than one ocean ecosystem driver is outside the norm simultaneously, in close spatial proximity or temporal succession (Leonard et al., 2014; Zscheischler et al., 2018). Major climate-related disasters often result from the compounding effect of multiple drivers and/or hazards. Such situations might arise, for example, when the drivers of one hazard (e.g. elevated temperature causing a marine heatwave) also cause other
relevant changes, such as decreased nutrient concentrations caused by increased thermal stratification and reduced supply of nutrient rich subsurface water to the surface. Compound events can severely impact marine ecosystems, especially when the hazards act synergistically. While MHWs may enhance mortality of some marine organisms, low productivity also threatens marine ecosystems that rely on phytoplankton as the base of their food web (Cavole et al., 2016). In recent years, interest in compound events has evolved into a rapidly growing research field (Zscheischler et al., 2020). However, most studies so far
focus on compound events over land (Ridder et al., in press). Few have addressed compound events in the ocean (Collins et al., 2019).

This lack of knowledge is of concern as MHWs often coincide with large anomalies in surface chlorophyll concentrations (Fig. 1). One of the most prominent example of a compound event is 'The Blob' in the Northeast Pacific. Between 2013 and 2015, the Northeast Pacific experienced the most intense and longest-lasting MHW ever recorded, with maximum surface tem-
perature anomalies of more than 5˚C (Fig. 1a) lasting for more than 350 days (Di Lorenzo and Mantua, 2016; Laufkötter et al., 2020). At its initiation, the MHW coincided with large negative anomalies in phytoplankton production along the California Current (∼28˚–48˚N) because of below-averaged strength in coastal upwelling, resulting in low chlorophyll levels through-out spring and summer (Leising et al., 2015). Later in 2014-2015, high temperature and low chlorophyll concentrations were observed further south in the eastern equatorial Pacific. The compound high temperature and low chlorophyll/nutrient event
had severe consequences for marine life (Cavole et al., 2016). Ecosystem impacts included low primary productivity (Whit-ney, 2015), extreme mortality and reproductive failure of sea birds (Jones et al., 2018; Piatt et al., 2020), mass strandings of whales in the western Golf of Alaska and of sea lions in California, and changes in species distribution in favor of warm-water species (Cavole et al., 2016; Cheung and Frölicher, 2020). These changes in biomass and species distribution further impacted socio-economically important fisheries (Cheung and Frölicher, 2020). However, not all MHWs coincided with low chlorophyll
events in the past (Fig. 1). During the Blob, for example, chlorophyll anomalies were positive in some locations (e.g. Bering Sea), exceeding on average the 80th percentile of their distribution, whereas along the Northwestern coast of North America and in the equatorial Pacific, chlorophyll anomalies fell on average below their 5th percentile.

Previous studies have identified drivers of MHWs (e.g. Holbrook et al. (2019)) and of chlorophyll variability (Boyce et al., 2010; McClain, 2009; Wilson and Adamec, 2002) separately, but it is currently unknown what the underlying drivers of
compound MHWs and low-chlorophyll (LChl) events are (Frölicher, 2019). Global warming is the dominant driver of long-term changes in MHW frequency (Frölicher et al., 2018; Oliver, 2019). Yet, natural variability of the climate system also creates





situations that favor the occurrence of extreme events. Recent MHWs have been linked to various large-scale modes of climate variability (Holbrook et al., 2019). These climate modes favor or suppress the occurrence of MHWs by modulating the local conditions. Bond et al. (2015) attributed the development of the Blob to an unusually strong and persistent weather pattern,

featuring much higher than normal sea level pressure over the Gulf of Alaska. These sea level pressure anomalies resulted from the strengthened Victoria Mode of variability in the northeast Pacific, which was forced by the atmosphere through the North Pacific Oscillation (Tseng et al., 2017). Reduced circulation in the North Pacific Subtropical Gyre suppressed the heat loss from the ocean to the atmosphere and caused relatively weak cold advection in the upper ocean (Leising et al., 2015). The resulting warming in the northeast Pacific is thought to have acted as a precursor to the development of the 2015/16

El Niño (Di Lorenzo and Mantua, 2016), which further enhanced the Blob (Tseng et al., 2017). Oceanic and atmospheric teleconnections associated with large-scale climate modes can also modulate the occurrence of MHWs in distant regions. For example, the extraordinary 2010–2011 La Niña remotely strenghtened and shifted the poleward-flowing Leeuwin Current along the western coast of Australia to the south. As a result the southwestern coast of Australia experienced anomalous warm waters in 2011 (Feng et al., 2013). Climate variability may also cause low productivity events, since large-scale climate modes

affect nutrient concentrations and primary production of phytophankton (Behrenfeld et al., 2001, 2006; Racault et al., 2017; Rousseaux and Gregg, 2014) at the surface via, for example, changes in mixed-layer depth and upwelling strength. Therefore, climate modes are potentially modulating the occurrence of compound MHW and LChl events and may be used to predict such events.

In this study, we provide a first characterization of compound MHWs and LChl events using satellite-based observations. We

first quantify the intensity and duration over time and space of MHWs and LChl events separately, before identifying hotspots and characterizing the temporal distribution of compound MHW and LChl events over the past decades. Finally, we investigate the modulation of their frequency by large-scale modes of climate variability.

## 2   Methods

### 2.1   Observation-based data

To identify and characterize compound MHW and LChl events, we use satellite-derived sea surface temperature (SST) and chlorophyll concentration data. For SST, we use NOAA's daily high-resolution Optimum Interpolation SST (OISST) analysis product with a spatial grid resolution of 0.25° (Banzon et al., 2016; Reynolds et al., 2007). This dataset provides a high quality daily global record of surface ocean temperature observations obtained from satellites, ships, buoys, and Argo floats on a regular grid. Its main input is infrared satellite data from the Advanced Very High Resolution Radiometer with high temporal-spatial

coverage beginning in late 1981 to the present. Any large-scale satellite biases relative to in-situ data from ships and buoys are corrected and any gaps are filled in by interpolation. For chlorophyll, we use outputs from the NASA Ocean Biogeochemical Model (NOBM.R2020.1 version) (Gregg and Rousseaux, 2017), which provides assimilated daily data for mean chlorophyll concentration within the mixed layer. This comprehensive ocean biogeochemical model, coupled to a global ocean circulation and radiative model (Gregg and Casey, 2007), assimilates satellite ocean chlorophyll data from the Sea-viewing Wide Field of





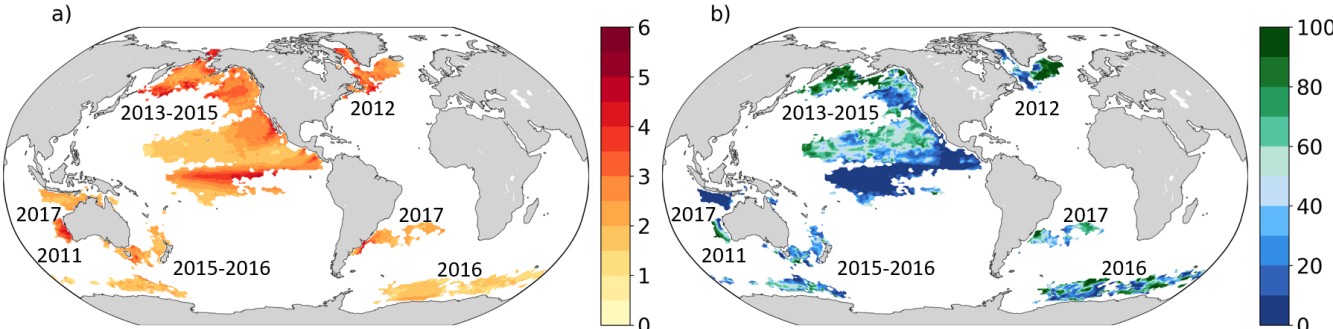

**Figure 1.** Recent (a) prominent large-scale marine heatwaves (MHWs) for which impacts have been documented, and (b) associated changes in surface chlorophyll concentrations ([chl]). We use sea surface temperature (SST) and [chl] deseasonalized anomalies. MHWs are defined when SST anomalies exceed their 99.5th percentile locally and numbers indicate the years of the MHW occurrences. (a) SST anomaly averaged over the MHWs duration. (b) Percentile associated with the mean [chl] anomaly averaged over the duration of the MHW, compared to the local empirical distribution of [chl] daily anomalies from 1998 to 2018. The MHWs extent is taken from Laufkötter et al. (2020) and corresponds to a spatiotemporally continuous area where each grid cell exhibits daily temperature anomalies above the 99.5th percentile.

View Sensor (SeaWiFs), the Moderate Resolution Imaging Specroradiometer (MODIS)-Aqua, and the Visible Infrared Imaging Radiometer Suite (VIIRS). The model spans the domain from 84°S to 72°N in increments of 1.25° longitude by 2/3° latitude, including only open ocean areas where bottom depth exceeds 200 m. Temporal coverage extends from 1998 to 2018.

Prior to any analysis, the high resolution SST dataset is regridded onto the lower resolution chlorophyll dataset for the period from 1998 until 2018 so that the length of the SST dataset corresponds to the length of the chlorophyll dataset. As chlorophyll concentration is close to or equal to zero during winter in the polar regions when solar radiation is near zero, we removed all days during which a particular grid cell receives no solar radiation thereby focusing on the growing season. The daily shortwave radiation data was obtained from the Modern-Era Retrospective analysis for Research and Applications version 2 (Gelaro and Coauthors, 2017).

## 2.2 Analysis

### 2.2.1 Definition of extreme and compound extreme events

We first computed anomalies by subtracting the mean daily seasonal cycle from the SST and chlorophyll data, respectively. MHWs and LChl events may therefore occur in any season, if solar radiation is non-zero. The mean seasonal cycle has been smoothed using a 30-day running mean to remove noise on a daily scale associated with the relatively short data record. We also used a 7-day and 14-day running mean for smoothing, but the main results are not sensitive to this choice.

Figure 2 illustrates our definition of univariate and compound extreme events. We define MHWs (i.e., hot temperature extremes) as events when the daily SST anomaly exceeds its local 90th percentile (Fig. A1a). Following this definition, MHWs can be as short as one day and extend over only one grid cell. Respectively, we define low-chlorophyll (LChl) events as days



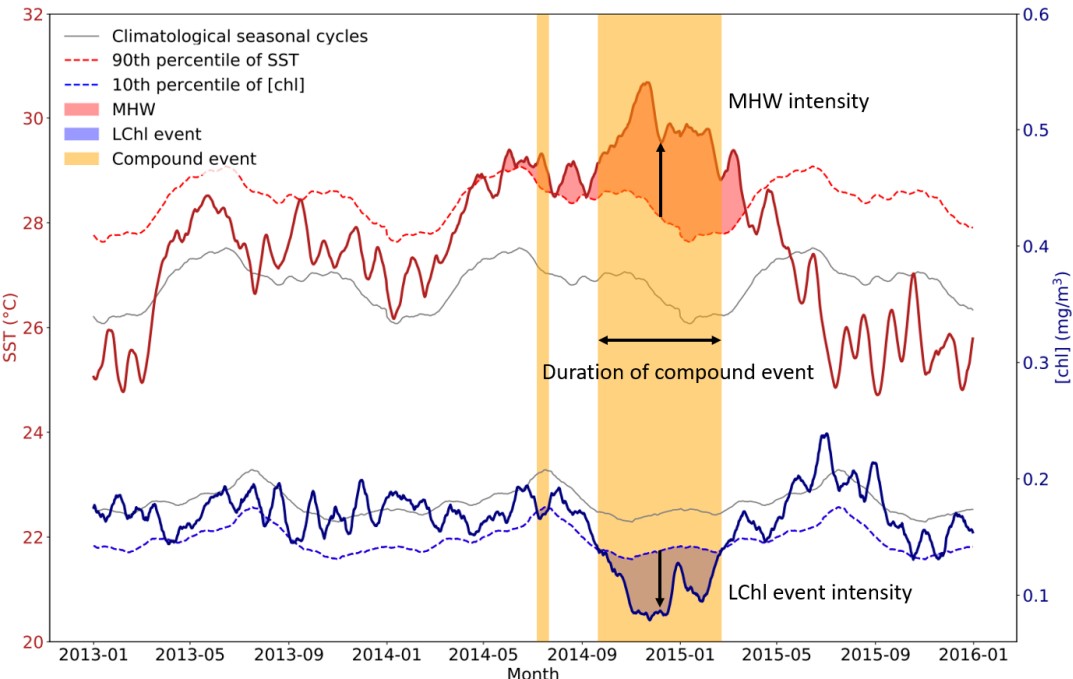

**Figure 2.** Schematic figure illustrating the definition of MHWs, LChl events and compound MHW and LChl events. Time series of SST and chlorophyll concentration are extracted from 2013 to 2015 at 0°N and 155°E, and smoothed with a 14-day running mean. A MHW occurs (red shaded area) when the SST (bold red line) exceeds its 90th percentile (dashed red line). A LChl event (blue shaded area) occurs when the surface chlorophyll concentration (bold blue line) is below its 10th percentile (dashed blue line). Yellow bands indicate the occurrence of compound MHW and LChl events.

when the anomaly in the chlorophyll concentration is below its local 10th percentile (Fig. A1b). We have chosen to focus here on rather moderate extremes, as defined with the 90th and the 10th percentile, because they provide sufficiently large sample
110 sizes for robust statistical assessments given the relatively short chlorophyll record from 1998 to 2018.

Compound MHW and LChl events are defined when both extreme hot temperatures and low chlorophyll conditions co-occur in time and space (yellow bands in Fig. 2). For simplicity we refer to them as "compound events". If MHWs and LChl events were independent, we would expect compound events to occur at a frequency (f) equal to the product of their univariate frequencies at each grid cell, that is $10\% * 10\% = 1\%$. The likelihood multiplication factor (LMF) of compound events is
115 defined as the ratio of the observed frequency of events to their expected frequency under the assumption of independence (Zscheischler and Seneviratne, 2017).

$$LMF = \frac{f(\text{Compound MHW and LChl event})}{f(\text{MHW}) * f(\text{LChl event})} = \frac{\text{Compound MHW and LChl event frequency (\%)}}{1\%} \quad (1)$$

Since the latter equals to 1% in our case, compound event frequency (in %) and the likelihood multiplication factor are equivalent. Thus, compound events occur particularly often at grid cells where their frequency exceeds 1% of all days.





### 2.2.2 Metrics for characterizing univariate extremes and compound extremes

We compute the duration, intensity, and frequency of extreme events. The duration of an univariate and compound extreme event corresponds to the number of days the event lasted without interruption (Fig. 2). Since the distribution of event duration is heavily skewed, we present the 90th percentile of the duration distribution in maps. The intensity of a MHW is defined as its mean SST exceedance anomaly over the duration of the event (Fig. 2). It corresponds to the difference between the mean SST anomaly over all MHW days of an event and the 90th percentile of SST anomalies. The intensity of a LChl event is defined as its mean chlorophyll exceedance anomaly, which corresponds to the difference between the mean chlorophyll anomaly over the duration of the event and the 10th percentile of chlorophyll anomalies. The intensity of a compound event is characterized by both the mean SST exceedance anomaly and the mean chlorophyll exceedance anomaly over the duration of the event in a bivariate plane. Finally, the frequency of an event is the number of event days over the total number of days, expressed as a percentage.

### 2.2.3 Attributing extreme and compound extreme occurrence to large-scale modes of climate variability

Large-scale modes of interannual to decadal climate variability may strongly modify the occurrence of MHWs, LChl events, and compound events. Holbrook et al. (2019) established an analytical framework to identify regions where statistically significant relationships exist between surface MHW occurrence and large-scale climate modes. Following the approach by Holbrook et al. (2019) for MHWs, we compute the frequency of MHWs, LChl events, and compound events during both positive and negative phases of the most relevant large-scale climate modes. These climate modes are: the El Niño Southern Oscillation (ENSO), the El Niño Modoki, the Indian Ocean Dipole, the North Atlantic Oscillation (NAO), and the Antarctic Oscillation (AAO). In the following, we briefly describe the individual modes and where we obtained the necessary data. The Niño-3.4 index indicates the state of ENSO and corresponds to the area-averaged SST anomaly in the equatorial central Pacific from 5°S-5°N and 170-120°W, relative to the mean SST over this area (www.esrl.noaa.gov/psd/gcos_wgsp/Timeseries/Nino34/). The El Niño Modoki is equivalent to the Central Pacific ENSO. It is estimated using the El Niño Modoki Index (EMI), which is based on the difference between SST anomalies in the central equatorial Pacific and the averaged eastern and western Pacific SST anomalies (http://www.jamstec.go.jp/frsgc/research/d1/iod/modoki_home.html.en). The Dipole Mode Index (DMI) measures the strength of the Indian Ocean Dipole. It is based on the difference between SST anomalies in the western equatorial Indian Ocean (50°E–70°E and 10°S–10°N) and in the south-eastern equatorial Indian Ocean (90°E–110°E and 10°S–0°N) (http://www.jamstec.go.jp/frsgc/research/d1/iod/e/iod/dipole_mode_index.html). The NAO index is constructed by projecting the daily 500mb height anomalies over the Northern Hemisphere onto the loading pattern of the North Atlantic Oscillation. The latter oscillation of atmospheric pressure anomalies consists of a north-south dipole with one center located over Greenland and the other center of opposite sign over the North Atlantic between 35°N and 40°N (www.cpc.ncep.noaa.gov/products/precip/CWlink/pna/nao.shtml). Finally, the AAO index is constructed by projecting daily 700mb height anomalies poleward of 20°S onto the loading pattern of the Antarctic Oscillation (https://www.cpc.ncep.noaa.gov/products/precip/CWlink/daily_ao_index/aao/monthly.aao.index.b79.current.ascii.table). In total we





consider the impact of ten different climate phases (positive and negative phases of five climate modes) on the frequency of univariate extreme and compound extreme events.

For each climate mode, we define positive, negative, and neutral phases based on their index values. We consider all days associated with the 50% lowest absolute values of the climate index to be in a neutral phase. Days associated with its most positive or negative values are respectively in a positive or negative phase. To estimate whether a climate phase has a discernible effect on local univariate and compound extreme events, we compare the frequency of extreme event days over each climate phase to their frequency over the complete 1998-2018 period at each grid cell. To ensure these frequency changes are

statistically significant, we shuffle the temporal order of each climate index and recompute the frequency change in extreme event days a thousand times for each grid point. If the observed frequency increase/decrease during a particular climate phase is higher/lower than 95% of the shuffled cases, we consider the association of that climate phase with a change frequency of extreme events significant at that grid point ($\alpha = 0.1$). We then also report significant associations for those climate modes that lead to the largest increase in extreme events for each location.

## 3    Results

We first assess the intensity and duration of MHWs and low-chlorophyll events separately (section 3.1), before we analyze spatial and temporal distribution of compound extreme events (section 3.2-3.3) and their drivers (section 3.4).

### 3.1    Marine heatwaves and low-chlorophyll events

The strongest MHWs with mean temperatures of up to 2°C above the 90th percentile are observed in high latitudes in regions

with high temperature variability (Oliver et al., 2018; Holbrook et al., 2019; Deser et al., 2010), such as the Western Boundary Currents and the Agulhas Current and Return Current, but also in the eastern equatorial Pacific (Fig. 3a). Less intense MHWs ($< 0.3$°C) occur in the western part of the subtropical gyres, the northern Indian Ocean and south of 45°S. The mean intensity pattern of low chlorophyll events broadly resembles the MHW intensity map, but with distinct differences (Fig. 3b). The most intense (up to -0.2 mg m$^{-3}$) LChl events are located at high latitudes, especially in the seasonally varying sea ice region of the

Southern Ocean, the North Atlantic, and the North Pacific. LChl events are also intense in the equatorial Ocean, but in contrast to MHWs, the mean intensity is not as pronounced in the eastern equatorial Pacific. Less intense chlorophyll extremes ($< -0.02$ mg m$^{-3}$) are generally found in the tropics and mid-latitudes, similar as for MHWs.

    The spatial distribution of the 90th percentile of MHW and LChl event durations is shown in Fig. 3c,d. Globally, LChl events last as long as MHWs. The 90th percentile of their duration is 12 days in both cases. Particularly long MHWs ($> 20$

days) occur in the eastern equatorial Pacific, where prolonged El Niño conditions may sustain positive SST anomalies for a few months and occasionally for up to two years (Fig. 3c). Long MHWs ($> 30$ days) are also observed in the seasonally varying sea ice region of the Southern Ocean and the Northeastern Pacific. Short MHWs ($< 5$ days) are found in the western part of the subtropical gyres, where the intensity of MHWs is also weak. Similar to MHWs, long LChl events are observed in high latitudes, in particular in the Southern Ocean around Antarctica, where ten percent of events last longer than a month.





Maximum durations are found in the Weddell Sea, where LChl events last up to 130 days. Similarly to MHWs, short LChl events are located in the mid-latitudes. In contrast to MHWs, low chlorophyll extremes last longest in the equatorial Atlantic and in the center of the equatorial Pacific, where El Niño oscillations may lead to zonal shifts of warm surface waters and high variability in phytoplankton growth conditions (Fig. 3d).

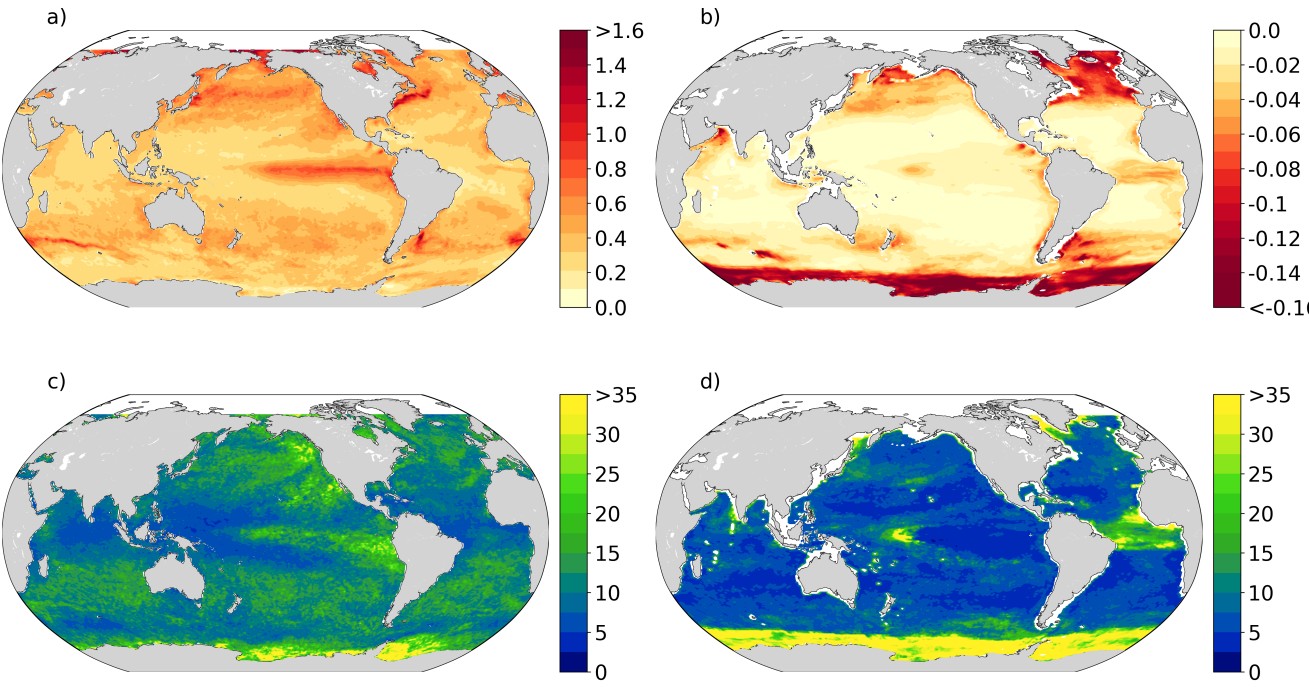

**Figure 3.** Observed marine heatwave and low chlorophyll extreme event characteristics averaged over the 1998-2018 period. (a,b) Mean intensity of MHWs (°C) and LChl events (mg m$^{-3}$). (c,d) 90th percentile of the duration of MHWs and LChl events in days.

## 3.2 Compound marine heatwaves and low-chlorophyll events

MHWs and LChl events often occur simultaneously. Indeed, the frequency of compound MHW and LChl events exceeds 1% in most of the global ocean (over 80% of the area), indicating that MHWs and LChl co-occur more often than if variations in SST and chlorophyll anomalies were independent (Fig. 4a). Globally, the average frequency of compound event days is 1.65%. Compound events are especially frequent ($> 2\%$ of all days) in the equatorial Pacific, along the boundaries of the subtropical gyres, in the Arabian Sea and around Antarctica. On the contrary, compound events occur less than 1% of days in the North
Atlantic and in the North Pacific, in the Indian Ocean south of 15°S, and in the Southern Ocean between 40°S and 60°S.



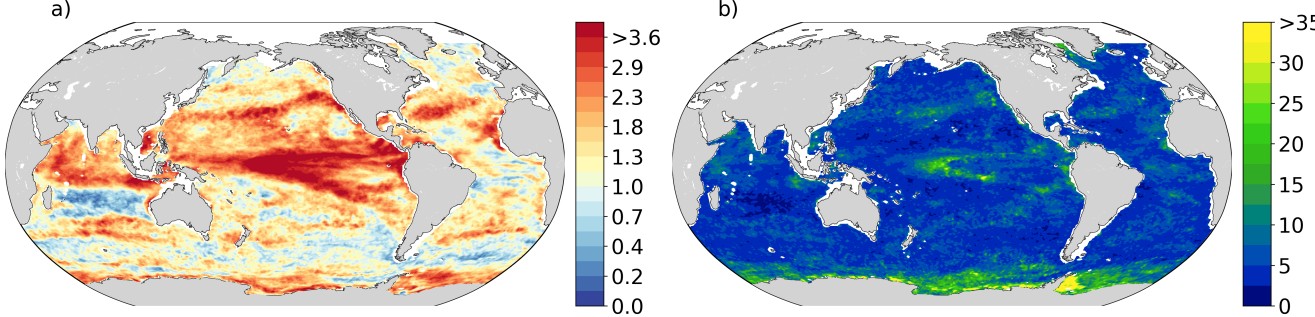

**Figure 4.** Observed frequency and duration of compound marine heatwaves and low chlorophyll extremes averaged over the 1998-2018 period. (a) Frequency of compound MHW and LChl event days (%). Note the nonlinear scale. Here, the frequency is equivalent to the likelihood multiplication factor. (b) 90th percentile of the duration of compound events.

Hotspots of compound MHWs and LChl events are typically located in regions where SST and chlorophyll anomalies are strongly negatively correlated (Fig. 5a), indicating that the overall dependence between SST and chlorophyll is not fundamentally different from the dependence in the tails of the distributions. The highest frequencies of compound event days (> 6% of all days) occur at grid points for which this correlation coefficient is especially negative (r < -0.5; Fig. 5b). The most frequent events (> 7% of all days) occur in the center of the equatorial Pacific. Here, the negative correlation between SST and chlorophyll anomalies is much lower than -0.5 (Fig. 5a). Grid points with positive correlations between SST and chlorophyll anomalies (r > 0.2) tend to have low frequencies of compound event days (< 1% of all days). Overall, there is a strong relationship (r = -0.74) between the occurrence of compound events and the correlation coefficient between SST and chlorophyll anomalies (Fig. 5b).

The frequency pattern of compound MHW and LChl events shown in Fig. 4a also resembles to some extent the observed temperature and chlorophyll concentrations during the most recent prominent large-scale marine heatwaves (Figure 1). Chlorophyll concentrations were exceptionally low in the eastern equatorial Pacific during the 2013-2015 Northeast Pacific MHW and in the Indonesian Sea during the 2017 MHW. These are regions where the compound event frequencies are very high (Fig. 4a). Chlorophyll concentrations were normal or high during the 2012 Northwest Atlantic MHW and during the 2016 MHW in the Southern Ocean, regions where the compound event frequency is also low. There are exceptions however. Some exceptions also occurred, such as in the northern subtropical Pacific gyre where chlorophyll concentrations were locally high during the 2013-2015 MHW, even though compound MHW and LChl events are relatively frequent (> 1.8% of all days).

Next, we assess the duration of compound events. On average, compound MHW and LChl events are half as long as univariate extremes; the 90th percentile of their duration is 6 days, against 12 days for MHWs and LChl events (Fig. 4b vs. Fig. 3c, d). The longest compound events (> 15 days) occur in regions of longest MHWs or LChl events, i.e., in the center of the equatorial Pacific and in the seasonally varying sea ice region of the Southern Ocean (Fig. 4b). In the Weddell Sea, 10% of compound events last longer than a month. Long compound events (> 10 days) also occur along the boundaries of the subtropical gyres





in the North Pacific and in the Arabian Sea. The shortest compound events occur in the western part of the subtropical gyres and, in general, in the extra-tropics.

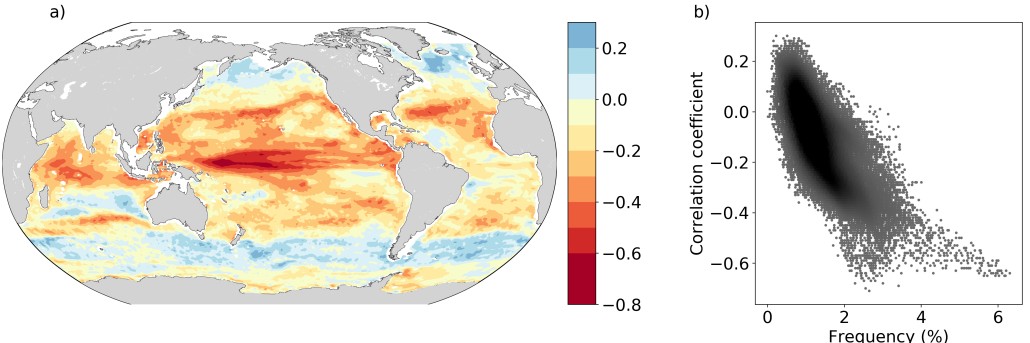

**Figure 5.** Correlation coefficient between SST and chlorophyll anomalies over the 1998-2018 period and its relationship to the occurrence of compound MHWs and LChl events. (a) Linear correlation coefficient between SST and chlorophyll anomalies. (b) Scatter plot of this correlation coefficient against the frequency of compound event days over the global ocean (see Fig. 4a).

Assessing the intensity of compound events is not as straightforward as assessing their frequency or duration, since they involve two variables. Figure 6 illustrates the joint 90th/10th percentile threshold exceedance anomalies of SST and chlorophyll anomalies averaged over all compound events at a grid point. These joint exceedance anomalies are generally low over most of the low- to mid-latitude ocean (green colors in Fig. 6). High exceedance anomalies are reached in regions exhibiting the most intense MHWs and LChl events (see Fig. 3a, b). Specifically, compound events with particularly warm SST (yellow and light

pink colors in Fig. 6) occur in the eastern equatorial Pacific, while compound events with particularly low chlorophyll (purple colors in Fig. 6) occur in the seasonally varying sea ice region of the Southern Ocean, in parts of the North Atlantic, and in the equatorial Atlantic. Intense compound events characterized by both extremely warm SST and low chlorophyll concentration (pink colors) occur at high northern latitudes, in eastern boundary upwelling regions such as the Canary, Humboldt and California upwelling systems, in the western boundary currents of the Atlantic, and in the center of the equatorial Pacific.




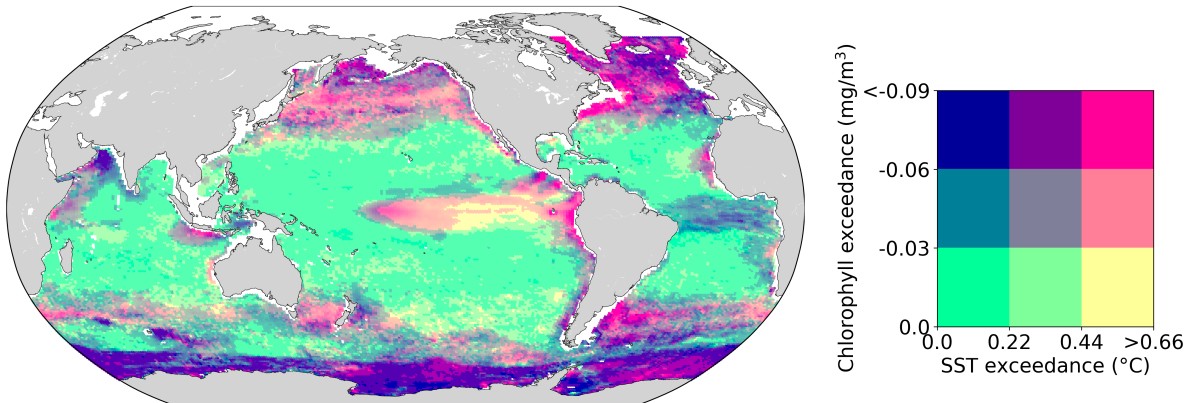

**Figure 6.** Intensity of compound marine heatwaves and low-chlorophyll events over the 1998-2018 period. Mean exceedance anomalies above the 90th and 10th percentile threshold of SST (°C) and chlorophyll (mg m$^{-3}$) anomalies, respectively, during compound events.

### 230  3.3  Distribution of marine heatwaves, low-chlorophyll and compound events over time

Next, we assess occurrences of MHWs, low-chlorophyll and compound events on seasonal to inter-annual time scales.

### 3.3.1  Seasonal time scale

At the global scale, MHWs predominantly occur between December and February, when their frequency exceeds 12% of days on average (not shown). Fewer MHWs occur between August and October ($< 9\%$ of all days). LChl events mostly occur
between November and January, whereas they are less common during the rest of the year ($< 10\%$ of all days). Consequently, most compound events occur in December and in January, when the probability of occurrence is high for both variables.

The seasonal occurrences of extreme events strongly varies with latitude. MHWs occur all year long at low and mid-latitudes (Fig. 7a). In high latitudes, MHWs mostly occur in summer, especially in the Southern Ocean where, on average, more than 14% of austral summer days are affected by a MHW, while there are almost no MHW days in austral winter and spring
south of 55°S. The temporal distribution over the year of LChl events is more heterogeneous across latitudes (Fig. 7b). While these events occur throughout the year along the equator (about 10% of days correspond to LChl event), they seem to follow the onset of the spring bloom in mid latitudes. This onset varies over the spring, resulting in higher chlorophyll variability in spring, which may explain why LChl events occur more frequently in this season. In high latitudes, LChl events mostly occur in summer, especially in the Southern Ocean. As a result of the temporal distributions in MHW and LChl occurrences, compound
events predominantly follow the distribution of LChl events over the year (Fig. 7c). At low latitudes, compound events occur at a similar frequency throughout the year. They mostly occur in spring at mid latitudes and in summer at high latitudes, with especially high frequency ($> 3\%$ of days) in the Southern Ocean in austral summer.





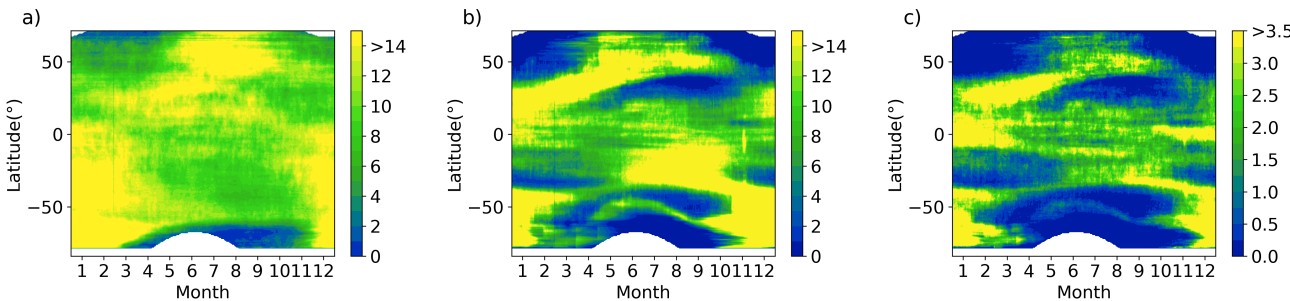

**Figure 7.** Seasonal cycle of MHWs, LChl events and compound events over the 1998-2018 period. Frequency of (a) MHW, (b) LChl and (c) compound event days (%) as a function of latitude and day of the year.

### 3.3.2 Interannual time scale

The occurrence of extreme events also varies at the interannual time scale from 1998 to 2018 (Fig. 8). In 1998, 2010 and
2015-2016, the frequency of MHW days exceeded 15% on average over the global ocean (Fig. 8a and Oliver et al. (2018)).
The mean frequency of MHWs is positively correlated with time series of the El Niño 3.4 index ($r = 0.54$). As indicated by the
red bands, these years were all characterized by strong El Niño events. These years were also characterized by longer MHWs,
especially in 2015 when MHWs lasted more than 30 days on average (not shown). In contrast, the occurrence probability and
duration of MHWs is reduced during La Niña events (blue bands in Fig. 8a). LChl events also vary over the years, but to
a smaller extent than MHWs. The frequency of LChl days strongly increased during the 2015-2016 El Niño event to up to
15% on average over the global ocean, but other strong El Niño events had almost no impact on the frequency of LChl events
(Fig. 8b), also exemplified by the low correlation coefficient of $r = 0.23$ between LChl events and the El Niño 3.4 index. Most
of the interannual distribution of compound events seems to be explained by the interannual distribution of MHWs, as LChl
events are relatively uniformly distributed over the years. Compound events occurred most frequently ($> 2.5\%$ of all days) in
the global ocean in 1998 and 2015-2016 (Fig. 8c), years which are characterized by strong El Niño events. Their frequency is
more positively correlated with El Niño 3.4 ($r = 0.42$) than the frequency of LChl events but slightly less than the frequency
of MHWs.



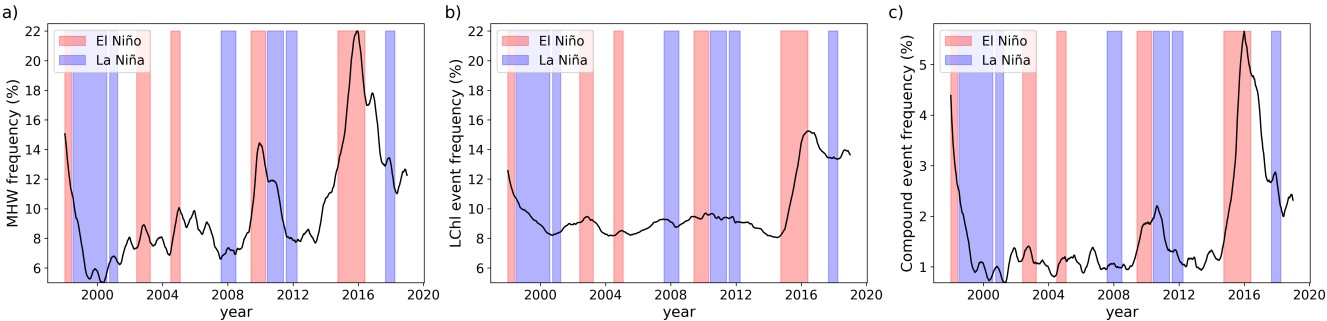

**Figure 8.** Global mean frequency of MHWs (a), LChl events (b) and compound events (c) over time. Daily time series are smoothed with a 1-year running mean to highlight interannual variability. Red and blue shadings indicate the occurrence of El Niño and La Niña events, respectively. These events occur when the El Niño 3.4 index exceeds 0.4 or is lower than -0.4, respectively, for at least 6 months. Note the different y-axes scales.

## 3.4 The role of natural internal climate variability

To improve our understanding of when and where compound MHW and LChl events occur, we identify the large-scale modes of internal climate variability that are associated with compound events locally. We compute the frequency of compound event days during the positive and negative phases of five different climate modes (see Methods). Figure 9 presents the frequency change in compound event days during these climate modes compared to the whole 1998-2018 period.

Overall, the relationship between compound event occurrence and climate modes are rather complex, but there are clear patterns emerging that are consistent with the well-known SST or atmospheric pressure patterns during these modes. The positive phase of ENSO (i.e., El Niño events) is associated with increased frequency of compound events in the central and eastern equatorial Pacific ($> 300\%$) (Fig. 9a). In 2015, an El Niño event was indeed associated with a compound MHW and LChl event (Fig. 1); chlorophyll anomalies reached extremely low values as the warming extended southward into the eastern equatorial Pacific due to the enhancement of the Blob by El Niño (Tseng et al., 2017). El Niño events are also associated with increased frequency of compound events in the Indian Ocean ($> 150\%$), the Pacific Sector of the Southern Ocean, and the California Current system, whereas they suppress compound events in the western Pacific (- 100%) and in the mid-latitudes (Fig. 9a). On the contrary, the negative phase of ENSO (i.e., La Niña events) is associated with higher frequency in the western Pacific ($> 150\%$) and in the Southern Ocean from 30°S to 50°S ($> 150\%$) and fewer compound events in the eastern equatorial Pacific and in the Indian Ocean (Fig. 9b). During the positive and negative phases of El Niño Modoki (EMI; Fig. 9c,d), the pattern of compound event frequency broadly resembles the pattern during positive and negative ENSO phases. However, compound events in the eastern equatorial Pacific and the Indian Ocean are less affected by EMI than by ENSO. The positive phase of the Indian Ocean Dipole (i.e., DMI+) is associated with higher frequency ($> 75\%$) in the Arabian Sea and reduced frequency ($< -75\%$) around the Maritime Continent (Fig. 9e). The positive phase of the North Atlantic Oscillation is associated with increased frequency of compound events in the North Atlantic mid-latitudes and in the northeastern Pacific, while it suppresses compound events in the North Atlantic high and low latitudes (Fig. 9g). Finally, the positive phase of the



Antarctic Oscillation is associated with higher frequency of compound events ($> 75\%$) in parts of the Southern Ocean, of the eastern Pacific, and of the eastern Indian Ocean (Fig. 9i).

    In general, the positive and negative phases of each climate mode are associated with opposite changes in the frequency of compound events. However, Fig. 9a,c,e,g,i are not exactly complementary to Fig. 9b,d,f,h,j, which partly reflects that the modes themselves are not perfectly complementary, e.g. there are asymmetries in the spatial structure, amplitude, duration and

time evolution of El Niño and La Niña (An and Jin, 2004; Dommenget et al., 2013; Okumura and Deser, 2010).





**Figure 9.** Frequency change in compound event days during positive and negative phases of several climate modes compared to their frequency over the complete 1998-2018 period (in %). Analyzed climate modes are: (a,b) El Niño Southern Oscillation, (c,d) El Niño Modoki, (e,f) Indian Ocean Dipole using its Dipole Mode Index, (g,h) North Atlantic Oscillation, and (i,j) Antarctic Oscillation. Grids cells where this frequency change is not statistically relevant remain white.





The climate mode associated with the largest frequency increase in compound event days varies over the global ocean (Fig. 10). ENSO is the main modulator of compound events occurrence in the eastern equatorial Pacific and in the northwestern part of the Indian Ocean, where El Niño events are associated with the highest frequency of compound event days from 1998 to 2018. El Niño Modoki leads to the greatest occurrence of compound events in some parts of the central equatorial Pacific and of the eastern equatorial Atlantic. The Indian Ocean Dipole is the main contributing climate mode to the occurrence of compound events days around Indonesia and in the subtropical Pacific. The North Atlantic Oscillation is the main modulator in the Gulf Stream region and in some parts of the northeastern Pacific. Finally, the Antarctic Oscillation is associated with the highest frequency of compound events in some parts of the Southern Ocean. Figure 10 is patchy in many areas, presumably due to the short 1998-2018 time period. The climate modes associated with the largest frequency increase in MHW days and in LChl event days, separately, are provided in the Appendix for reference (Fig. A2).

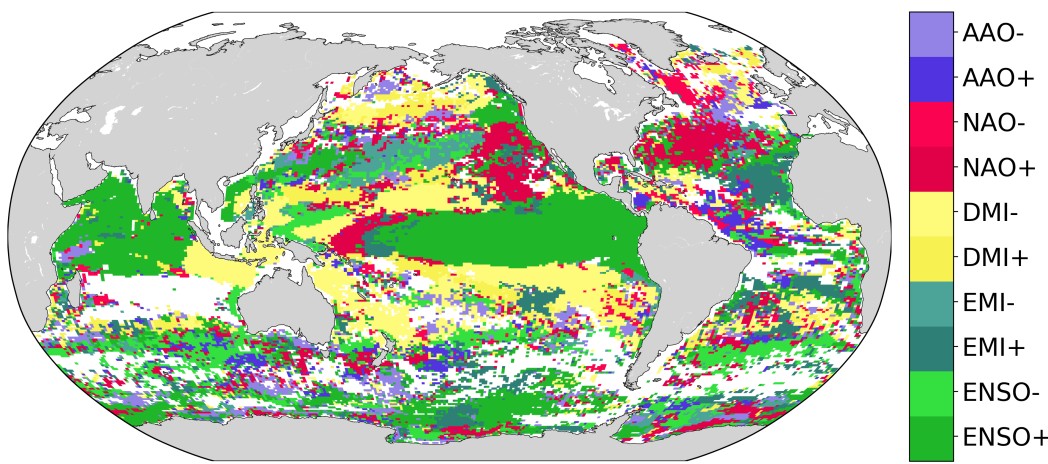

**Figure 10.** Phases of large-scale climate modes associated with the highest frequency of compound event days from 1998 to 2018. Regions for which the base frequency over the whole 1998-2018 period is <1% are marked white.

## 4    Discussion and Conclusion

In this study, we provide a first assessment of compound marine heatwave and low chlorophyll extreme events in the global ocean over the 1998-2018 period. We show that hotspots of compound MHW and LChl events can be found in the equatorial Pacific, along the boundaries of the subtropical gyres and in the Arabian Sea, regions where the sea surface temperature and chlorophyll anomalies are predominantly negatively correlated. Furthermore, we show that compound events mostly occur in summer in the high latitudes and throughout the year in the low to mid-latitudes, and that different large-scale modes of climate variability are associated with compound MHW and LChl events.





Our identified global pattern of compound event frequency in Figure 4a corresponds to some extent with the results of Hayashida et al. (2020), who concluded that the general response of chlorophyll to MHWs at specific sites depends on the

background surface nutrient concentration. They showed that during MHWs, shallower mixed layer depth and lower nitrate concentration exacerbate nutrient stress (Behrenfeld et al., 2006; Racault et al., 2017), resulting in lower chlorophyll concentration in nutrient-limited surface waters, whereas the relief of light limitation during MHWs leads to higher chlorophyll concentrations in nutrient-rich surface waters. The eastern equatorial Pacific is an exception, where, for example during El Ninõ events, the reduction in the upwelling of cold and nutrient-rich subsurface waters leads to lower chlorophyll concentra-

tion (Hayashida et al., 2020; Racault et al., 2017). The decrease in chlorophyll concentration may be exacerbated by a potential increase in grazing pressure in warmer waters in these eutrophic waters (Laufkötter et al., 2015; Kutlu, 2012). In contrast to the Hayashida et al. study, which investigates the general response of chlorophyll to MHWs at specific sites regardless whether the chlorophyll concentration is extreme or not, our analysis identifies regions at the global scale where both temperature and chlorophyll are extreme at the same time. Despite this difference, we also identify elevated compound event frequency in the

nutrient-limited surface waters of the low latitudes (including the eastern equatorial Pacific) and low compound event frequency in the nutrient-rich surface waters of the Southern Ocean (Fig. 4a). There are exceptions, however. Compound events are relatively frequent in the North Pacific, North Atlantic and around Antarctica even though the background nutrient concentration is relatively high in these regions. In addition, the frequency of compound events in the Indian Ocean is relatively low even though the surface nutrient concentrations are low there. This calls for additional process-oriented studies to identify the exact

physical and biogeochemical processes driving compound high temperature and low chlorophyll events.

While there is a growing understanding on how the occurrence of MHWs changes under several modes of internal climate variability (Holbrook et al., 2019), the modulation of LChl events and in particular of compound MHW and LChl events frequency is barely understood. We assessed changes in the frequency of compound events during the positive and negative phases of several climate modes. Even though statistical relationships do not necessarily indicate causal links, these changes

help predict the occurrence of compound events as a function of the oceanic region and the state of a climate mode. For example, compound event frequency is increased by up to 300% in the Pacific and Indian Oceans during El Niño events. We can therefore expect frequent compound events in these regions during upcoming El Niño events. The relationships we demonstrate between climate drivers and compound events resemble the relationships between climate drivers and MHWs shown in Holbrook et al. (2019) in the equatorial Pacific, in the Indian Ocean, and in the northern Atlantic. Therefore, the

state of the different large-scale climate modes can potentially be used to predict the occurrence of both MHWs and compound MHWs and LChl events in these regions. However, this is not the case in other regions (e.g. over the southern Pacific Ocean), where the occurrence pattern of MHWs and compound MHW and LChl events differ for the different large-scale modes of climate variability. This indicates again that other processes (see above) may affect chlorophyll concentrations in these regions or that MHWs are mostly modulated by climate modes that we omitted in our study (e.g. the Pacific Decadal Oscillation in

the northeastern Pacific) because our shorter period of analysis does not capture their variability. Given that compound events are strongly associated with several large-scale modes of climate variability, skillful multi-annual forecasts of the state of these climate modes may be used as an early warning system for the occurrence of compound events, and may therefore provide





critical information for fisheries management and adaptation interventions to reduce risks and impacts on marine organisms and ecosystems during such events (Holbrook et al., 2020).

Even though we consider our results robust, two potential uncertainties need to be discussed. First, our quantitative results are sensitive to the particular assumptions that need to be made during the statistical analysis (e.g. threshold value, fixed vs. moving baseline; Burger et al. (2020); Oliver et al. (2021)). We chose to use the 90th/10th percentile thresholds to have a relatively large number of compound events given the relatively short satellite record for chlorophyll. Choosing different thresholds led to qualitatively similar results. In addition, we use a fixed baseline climatology (i.e., the entire 1998-2018 satellite record).
Therefore, any long-term changes in sea surface temperature and chlorophyll affect the frequency of compound events over time. Because there is a gradual increase in mean sea surface temperature (SST), MHWs generally occur more often towards the end of the satellite record (Fig. 8a; Frölicher et al. (2018); Oliver et al. (2018); Laufkötter et al. (2020)). This is not the case for LChl events, as the long-term trend in mean chlorophyll concentrations is close to zero (Hammond et al., 2020; Rousseaux and Gregg, 2014). Consequently, a fixed baseline might not affect the occurrence of LChl events, but it might favor an increase
in the occurrence of compound MHW and LChl events over the satellite period along with the increase in mean SST.

    Second, whereas the satellite-derived temperature data has been validated extensively (Banzon et al., 2016; Huang et al., 2020; Reynolds et al., 2007) and used for many recent marine heatwave analysis (e.g. Hobday et al. (2018); Oliver et al. (2018); Frölicher et al. (2018); Laufkötter et al. (2020)), the satellite-derived chlorophyll estimates have not been extensively used to analyze extreme events. High solar zenith angles, clouds, aerosols and interorbital gaps can lead to a bias in the
chlorophyll (and temperature) data (Gregg et al., 2009). Furthermore, the data have to be merged over several weeks or even months to achieve true global representation. By assimilating satellite ocean color in the NASA Ocean Biogeochemical Model, we reduced some of these biases. Nevertheless, we note that our results need to be taken with caution, especially near the coasts and at high latitudes, where the chlorophyll estimates remain uncertain.

    Impacts of compound MHW and LChl events on marine organisms and ecosystems may be more severe than the impacts
from MHWs and LChl events individually. Even though little is known about the impacts of compound MHW and LChl events, many studies have documented the mostly strong negative effects of MHWs alone. It is assumed that marine species are particularly vulnerable to MHWs in the low latitudes, since these species already live at the upper thermal edge of their habitat (Smale et al., 2019). MHWs in the low latitudes also have critical impacts on foundation species such as corals, seagrass and kelp (Smale et al., 2019). In the high latitudes, where biological production is often light limited (McClain, 2009), MHWs may
be beneficial for some species as long as MWS are not very abrupt, prolonged and compounded with other stressors over time (Cavole et al., 2016; Walsh et al., 2018). On the other hand, low chlorophyll, when indicating lower net primary production, results in lower food supply in all oceanic regions with harmful effects on marine biology. While chlorophyll is not always correlated with phytoplankton biomass or net primary production, particularly in subtropical regions (Barbieux et al., 2018), it is still commonly used as a proxy for phytoplankton biomass or net primary production (e.g. Behrenfeld et al. (2005); Henson
et al. (2010)). We therefore assume that LChl events often exacerbate the impacts from MHWs. In addition, phytoplankton includes a diverse range of different species that may respond differently to MHWs. For example, both the phytoplankton and zooplankton community composition have changed from larger species to smaller species during the Northeast Pacific





2013-2015 MHW (Cavole et al., 2016), resulting in less energy available for the food web. While some species benefited from the compound MHW and LChl event (e.g. rockfish, subtropical copepods, tuna and orcas), the mortality of many other species

substantially increased (subarctic copepods, crabs and mussels, sea birds, seals, sea lions and whales). More research is needed to understand the effects of exceptional warming events combined with LChl levels, as marine ecosystems could suffer severe damage.

Earth system models project further surface warming and decreasing primary production in nutrient limited waters of the low-to-mid latitudes during the 21st century (Bopp et al., 2013; Kwiatkowski et al., 2020). Given these projected long-term

trends, we can expect more frequent compound events and increasing pressure on marine organisms and ecosystems over the next decades in these regions. We therefore encourage future work aimed at assessing the vulnerability, adaptability and resilience of marine ecosystems to these compound events.

Our results provide a first characterization of where and when compound MHW and LChl events might occur, and how these events are associated with large-scale modes of internal climate variability. Additional observational-based and modelling

studies are needed to identify the exact physical and biological drivers of such compound events in the ocean, their evolution with climate change and their impacts on marine ecosystems.

## Appendix A

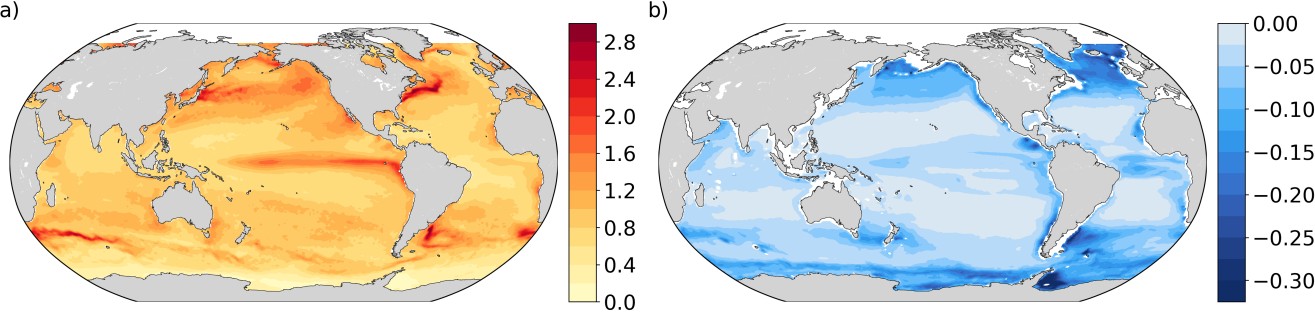

**Figure A1.** (a) 90th percentile of SST anomalies (°C) and (b) 10th percentile of chlorophyll anomalies (mg m$^{-3}$) from 1998 to 2010 at each grid cell.





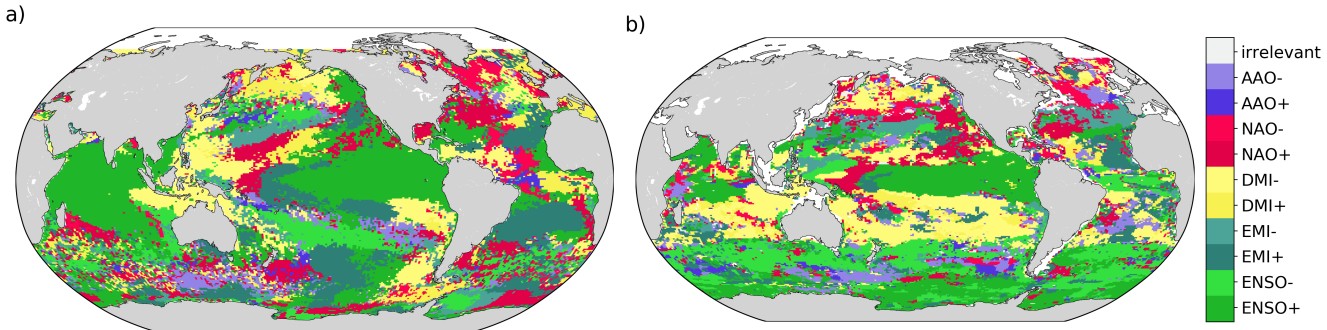

**Figure A2.** Phases of large-scale climate modes associated with the highest frequency of MHW days (a) and low-chlorophyll event days (b). Regions for which none of the climate modes reaches a significant change of the frequency (see Methods) are marked grey.

*Author contributions.* NLG, JZ and TLF designed the study, with substantial input from CL. NLG performed the analysis and wrote the initial draft of the manuscript. CSR provided the chlorophyll dataset. All authors discussed the analysis and results, and contributed to the
writing of the paper.

*Data availability.* The satellite SST data is available under https://www.ncdc.noaa.gov/oisst/data-access. The chlorophyll data assimilated by the NASA Ocean Biogeochemical Model is publicly available from 1998 to 2015 under https://disc.gsfc.nasa.gov/datasets/NOBM_DAY_R2017/summary. Cécile Rousseaux provided a pre-release of the chlorophyll data from 2016 to 2018, and this data is available upon request. The figures and analysis are available under following link on ZENODO: TO BE INSERTED

*Competing interests.* All authors declare no competing interests.

*Disclaimer.* The work reflects only the authors' view; the European Commission and their executive agency are not responsible for any use that may be made of the information the work contains.

*Acknowledgements.* NLG is funded by the Alfred Bretscher Fond. This research has been supported by the Swiss National Science Foundation (nos. PP00P2_170687, 174124, 179876), the European Union's Horizon 2020 research and innovation programme under grant agree-
ment No 820989 (project COMFORT, Our common future ocean in the Earth system — quantifying coupled cycles of carbon, oxygen, and



nutrients for determining and achieving safe operating spaces with respect to tipping points) and the Helmholtz Initiative and Networking Fund (Young Investigator Group COMPOUNDX, Grant Agreement VH-NG-1537). The authors acknowledge the European COST Action DAMOCLES (CA17109).



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
