# Peer review of "Compound high temperature and low chlorophyll extremes in the ocean over the satellite period"

_Biogeosciences, 2020_

## Referee Comment (RC1) · Anonymous Referee #1 · 10 Dec 2020

General Comments This manuscript uses satellite data to identify events in the ocean that occur at extremes of both SST and chlorophyll, "compound" events. This is an emerging field of study, particularly for the ocean and is important for understanding how our planet is changing under global warming. The paper is well written and suitable for publication. I have some suggestions for improving the manuscript.

Specific Comments

There has been a lot of work done, dating back to the 1980s, on coral bleaching events, which are largely temperature driven. It might be outside the scope of this paper, but it could be interesting to see if there are any similarities between the global distribution

and timing of coral bleaching events and the compound events described in this paper.

Avoid the use of terms like "high-resolution", "high quality" and "high temporal-spatial coverage" as they are not quantitative terms (section 2.1). For example, they say the SST dataset they use is "high-resolution" but also state that it has a resolution of 0.25°, which these days is not considered a high-spatial resolution for a SST dataset, just the opposite, it would be considered a rather coarse resolution, given that there are products available at < 1 km spatial resolution. However, I realize that "high resolution" is part of the AVHRR acronym, so clearly the term needs to be used in that context. Similarly, they frequently refer to the 1998-2018 time period as being "short" (i.e., line 103, 299). Obviously, this is the longest record of satellite chl we have ever had. When these statements are made they need to clarify exactly how the 20 years of data is not sufficient, if that is indeed the case.

How do the chlorophyll results of the NASA Ocean Biogeochemical Model compare to the ESA OC-CCI product? I am more familiar with that product being used when there is a need for a dataset spanning across all OC sensors.

Technical Comments

Section 2.2.3 It is difficult to keep track of all of these climate indices as written here – it might be easier to digest if this information was presented in a table.

Figure 2. Why is just the time period 2013-2015 shown? And why this location?

Figure 4 & 5. Since the data shown in Figure 5 is very relevant to that shown in Figure 4, I wanted to be able to look at them side by side, which is difficult since they are on separate pages. Consider merging these together into one plot.

All Figures. Label the color bars with the variable shown and the units.

Section 3.3.1 The discussion in the first paragraph about the global scale distribution is hard to follow, especially since the data is not shown. My first thought was that there must be a difference in behavior between hemispheres, and I was wondering if

they took that into account, and then that is brought up in the next paragraph with the discussion of Figure 7. I suggest removing the first paragraph entirely, and just focusing on the behavior as a function of latitude and day of year, which clearly shows that the MHWs are more prevalent in the winter (northern and southern).

Figure 10. It is impossible to distinguish between the four different colors of green, and the two different yellows. Is there even much difference in the distribution between the negative and positive phases? If there is it is impossible to tell on this figure. I suggest redoing with just 5 colors, one per index.

Be consistent in the use of acronyms. Acronyms are used for the climate modes in Figure 10, but in the text acronyms are not used. Same for LMF, acronym is defined but not always used.

---

## Referee Comment (RC2) · Monique Messié (Referee) · 10 Dec 2020

General comments:

The paper by Le Grix et al. is a global analysis of marine heatwaves (MHW), low-chlorophyll events (LChl), and most importantly, compound events defined as both occurring simultaneously. The authors characterize these events in terms of intensity, duration, and frequency, describe their spatial and temporal patterns (including seasonal cycle and interannual variability), and analyze their link with well-known climate indices. This is an excellent paper, well-written and easy to follow. The results are novel and this is a welcome study, particularly in a context where MHWs have been

extensively studied but their association with reduced oceanic productivity less so. I do have 2 concerns detailed below that should be very easy to address but may significantly impact the results and some of the paper's conclusions. The authors are certainly welcome to not apply these suggestions, in which case this choice needs to be carefully justified (including within the paper as I expect other readers will have similar concerns).

Specific comments:

One of the strengths of this paper is its continuity with the literature, particularly Holbrook et al. (2019) who analyzed MHWs. Intensity, duration and frequency are defined similarly, and the link with climate indices is conducted following the same method (contrasting event frequency in a positive or negative phase). Fig. 10 is even constructed similarly to Fig. 3b in Holbrook et al (2019), with most colors matching. This makes it easy to compare results from this paper with results from Holbrook et al., which is very good. There are 2 ways this continuity should be further improved in my opinion, for consistency but also because results may be significantly impacted.

First, I would recommend that a duration threshold be used, at least for MHW and LChl events. Holbrook et al. used 5 days, following recommendations by Hobday et al (2016). Currently, MHW, LChl, and compounds events can be as short as one day. This goes against previous recommendations by Hobday et al (2016) and their qualitative definition of a MHW as a "discrete *prolonged* anomalously warm water event". While no definition exists for LChl in the literature that I am aware of, it does make sense to use a similar definition. While including short-duration events can be justified (and using a given threshold does introduce a bias too), my concern is that the lack of threshold is also certainly the reason for the "heavily skewed" duration distribution mentioned l. 123 that required the use of the 90th percentile, rather than mean, for describing duration patterns. Based on the "heavily skewed" distribution for duration, I am concerned that most of the results (averaged frequency/intensity) are heavily skewed, too, towards short-duration events that can be considered to hardly qualify as MWH,

LChl or compound events. Fig. 3d and Fig. 4b suggest that very few LChl and compound events > 5 days may be found over large parts of the ocean (where the 90th percentile is below 5 days); this is OK and an interesting result in itself. I understand that retaining all events justifies the 1% threshold for compound events; however, this threshold could be re-calculated at each pixel as the percentage of time the pixel is in a MHW multiplied by the percentage of time the pixel is in a LChl. The results could then be displayed as not only frequency (as in Fig. 4a), but also LMF relative to this local threshold as defined by Eqn (1). If you need to retain the < 5-day events, this needs to be very carefully justified – in particular, does it makes sense to consider that 10% of days at any location belong to a MHW event, and (another) 10% to a LChl event? Some figures should also be added to clarify how the results are impacted by these short-duration events. In particular, do frequency/intensity patterns change when only retaining events > 5 days?.

Second, additional climate indices used by Holbrook et al. should be included, specifically the PDO and NPGO that both have strong footprints in the northern Pacific (Holbrook et al Fig. 3b). While these modes are decadal, they both display positive and negative phases during the 1998-2018 time period so there is no reason why they could not be analyzed. Considering how prevalent the PDO and NPGO are in the Pacific, including them makes sense and may change some of the Fig. 10 results (if Holbrook's results are any indication, I would expect the NPGO to replace the NAO in the north-eastern Pacific, and the PDO to replace the EMI in the north-western Pacific). The NPGO may not have enough negative values over 1998-2018 but the impact of the positive phase could be assessed at least – particularly if the positive/negative phases where compared to the neutral values, rather than the mean (which I believe makes more sense anyways as the mean can be skewed towards positive or negative events).

Minor comments and technical corrections:

l. 87-88: This is mentioned in the discussion, but it would be worth mentioning here that

daily satellite chlorophyll cannot be used for this analysis because the data coverage is too poor at the daily scale (notably due to clouds).

Fig. 2: Were time series smoothed using a 14-day running mean prior to MHW/LChl/compound event definition? the text only mention smoothing the daily seasonal cycle. If time series were not smoothed in the calculations, they should not be smoothed in the figure. If they were, please update the text.

l. 123: see comments above regarding the heavily skewed distribution. If you decide to retain short-duration events, at the minimum the distribution should be displayed (eg. box plot) for readers to understand how much of an impact short-duration events may have on the results.

l. 159: as a suggestion and as stated above, consider comparing the frequency of extreme event days over each climate phase to their frequency over the neutral phase, rather than over the complete 1998-2018 period. How did you attribute events spanning several phases between positive/neutral/negative, particularly events that might be long enough to span both positive and negative phases?

l. 185 "Similarly to MHWs": I would actually argue, based on Fig. 3c vs d, that MHW and LChl have almost exactly opposite duration patterns over most of the global ocean when excluding the Southern Ocean.

l. 190 "MHWs and LChl events often occur simultaneously": In addition to comparing the compound event frequency to the expected frequency, it would be useful to report the percentage of MHWs events that coincide with a LChl event, and the percentage of LChl events that coincide with a MHW event ("day(s)" could be used instead of "event(s)" in this sentence, not sure which would be most informative).

Fig. 4: consider using a linear scale, or at least displaying "logical" colorbar ticks (e.g. 0, 1, 2, 3). As it is the results are difficult to visualize.

l. 196-204: while the fact that compound events are located in regions where Chl/SST

are negatively correlated makes perfect sense, this is still a very interesting result and it was nicely demonstrated.

l. 210 "There are exceptions however. Some exceptions also occurred. . .": Should one of these 2 sentences be removed? Not sure what you meant here.

l. 217 "Long compound events (> 10 days)": did you mean "where 10% of events last longer than 10 days"?

l. 291-300: As acknowledged in the discussion, correlation does not equate causation. Please rephrase sentences such as "El Nino Modoki leads to the greatest occurrence. . .", "The Indian Ocean Dipole is the main contributing climate mode. . .", "The North Atlantic Oscillation is the main modulator. . .". The modes are associated with high compound frequency (as said in other sentences) but we don't know if they drive them.

l. 313 and 320: aren't these sentences contradictory? the first indicate that the eastern equatorial Pacific is an exception to the Hayashida rule, the second indicate that the eastern equatorial Pacific behaves as expected. It almost seems like you consider the eastern equatorial Pacific to be nutrient-rich l. 313 and nutrient-limited l. 320.

l. 321-323: any hypothesis as to what may be at play in these regions?

l. 337-338: this is not what I see when comparing Fig. 10 to Holbrook et al (2019, their Fig. 3b) for the Southern Ocean. Both figures highlight ENSO and the AAO/SAM, and both display a very complex picture. Did you refer to the patchiness and frequency of white pixels, indicating that no clear signal can be identified?

l. 340: see comment #2. Particularly for the PDO, there are both positive and negative phases during 1998-2018.

---

## Author Response (AR1)

**Compound high temperature and low chlorophyll extremes in the ocean over the satellite period Response to reviewers comments**

Natacha Le Grix, Jakob Zscheischler, Charlotte Laufkötter, Cécile S. Rousseaux, and Thomas L. Frölicher

January 25, 2021

**1 Response to Anonymous Referee 1**

**1.1 General Comments**

This manuscript uses satellite data to identify events in the ocean that occur at extremes of both SST and chlorophyll, "compound" events. This is an emerging field of study, particularly for the ocean and is important for understanding how our planet is changing under global warming. The paper is well written and suitable for publication. I have some suggestions for improving the manuscript.

We thank the reviewer for the positive and encouraging feedback.

**1.2 Specific Comments**

There has been a lot of work done, dating back to the 1980s, on coral bleaching events, which are largely temperature driven. It might be outside the scope of this paper, but it could be interesting to see if there are any similarities between the global distribution and timing of coral bleaching events and the compound events described in this paper.

We thank the reviewer for this interesting comment. Whereas a full assessment of the similarities between coral bleaching events and compound events is outside the scope of this paper, we added to the Discussion section:

*"These correspond to regions where the sea surface temperature and chlorophyll anomalies are predominantly negatively correlated, and also to regions where most of the warmwater corals are located and where coral bleaching events often occurred in the recent past (Hughes et al. 2018)."* (l. 313-315)

Concerning the timing of coral bleaching events, El Niño became a predominant trigger of mass bleaching in the 1980s, when global warming increased the thermal stress of

strong El Niño events (Hughes et al. 2018). Our study shows that El Niño is also associated with increased frequency of compound events in the eastern equatorial Pacific and in the Arabian Sea. Therefore, coral bleaching events may often co-occur with compound events in these regions during the positive phase of ENSO. The 2015-2016 El Niño event was indeed associated with both a compound event and a coral bleaching event in the equatorial Pacific (Fig. 1 of this paper and Fig. 3 of Hughes et al. 2018). However, in the past two decades, many additional regional-scale bleaching events have occurred outside of El Niño conditions (Hughes et al. 2018), since temperature thresholds for bleaching are increasingly exceeded throughout all ENSO phases due to global warming. Thus, similarities in the timing of coral bleaching events and of compound events may change over time with global warming.

We decided not to include the above paragraph in the manuscript, as it is outside the scope of our study.

Avoid the use of terms like "high-resolution", "high quality" and "high temporal-spatial coverage" as they are not quantitative terms (section 2.1). For example, they say the SST dataset they use is "high-resolution" but also state that it has a resolution of $0.25\circ$, which these days is not considered a high-spatial resolution for a SST dataset, just the opposite, it would be considered a rather coarse resolution, given that there are products available at $< 1$ km spatial resolution. However, I realize that "high resolution" is part of the AVHRR acronym, so clearly the term needs to be used in that context. Similarly, they frequently refer to the 1998-2018 time period as being "short" (i.e., line 103, 299). Obviously, this is the longest record of satellite chl we have ever had. When these statements are made they need to clarify exactly how the 20 years of data is not sufficient, if that is indeed the case.

We agree and have applied the suggested changes to the manuscript. We have changed *"NOAA's daily high-resolution Optimum Interpolation SST (OISST) analysis product"* to *"NOAA's daily Optimum Interpolation SST (OISST) analysis product"* (l. 83); *".. provides a high quality daily.. "* to *".. provides a daily.. "* (l.84); *".. with high temporal-spatial.."* to *"..with temporal.."* (l. 86); *"relatively short data record"* to *"21-year data record"* (l.114).

How do the chlorophyll results of the NASA Ocean Biogeochemical Model compare to the ESA OC-CCI product? I am more familiar with that product being used when there is a need for a dataset spanning across all OC sensors.

The ESA chlorophyll product combines the existing satellite ocean color sensors (SeaWIFS, MODIS and VIIRS) and corrects for differences between sensors such as band location differences, the fact that each of the satellites observes parts of the Earth at different times, or that each sensor has its own signal to noise ratio. In our manuscript, we use a product that assimilates ocean color satellite data from SeaWIFS, MODIS and VIIRS into the NASA Ocean Biogeochemical Model (NOBM). We added the following sentences to the Methods section where we describe the chlorophyll dataset:

*"NOBM takes care of differences between sensors and also provides complete coverage at a daily resolution, without the gaps that are intrinsic to satellite data due to clouds and high solar zenith angles. Its chlorophyll outputs have been validated against the NASA satellite products (Gregg and Rousseaux 2014)."* (l. 95 - 98)

Indeed, even though the chlorophyll outputs of the NASA Ocean Biogeochemical Model have not yet been compared to the ESA OC-CCI product, they have been validated against the NASA satellite products (Gregg and Rousseaux 2014) as we further explain:

*"The annual median chlorophyll is similar when computed using the satellite products or the NOBM products, although in the high latitudes, areas of high chlorophyll in the satellite products are reduced in the assimilation data. According to Gregg and Rousseaux 2014, these are artifacts of satellites sampling only the warmer, more sunlit months while the assimilation model produces information for all days of the year. In the North Indian Ocean, high chlorophyll due to seasonal aerosol obscuration in the satellite product is also reduced when assimilated. Trends in global mean chlorophyll are similar from 1998 to 2012 in both the satellite and assimilation products."* (l. 98-103)

**1.3   Technical Comments**

Section 2.2.3 It is difficult to keep track of all of these climate indices as written here – it might be easier to digest if this information was presented in a table.

We added a table in section 2.2.3 that summarizes the climate modes used in our study, their indices, and their associated acronyms. We also moved the exact definition of these climate indices into the Appendix to facilitate the reading of section 2.2.3.

Figure 2. Why is just the time period 2013-2015 shown? And why this location?

We extract a short time period at one grid cell to facilitate the visualization of MHWs, LChl events and compound events on a single graph. We chose a grid cell in the equatorial Pacific and a time period between 2013 to 2015 because we knew from Fig. 1 that during the Blob, a long MHW co-occurred with extremely low chlorophyll in this region.

Figure 4 and 5. Since the data shown in Figure 5 is very relevant to that shown in Figure 4, I wanted to be able to look at them side by side, which is difficult since they are on separate pages. Consider merging these together into one plot.

We like this suggestion and merged the two figures. Thank you.

All Figures. Label the color bars with the variable shown and the units.

Variable and units were added to each figure.

Section 3.3.1 The discussion in the first paragraph about the global scale distribution is hard to follow, especially since the data is not shown. My first thought was that there must be a difference in behavior between hemispheres, and I was wondering if they took that into account, and then that is brought up in the next paragraph with the discussion of Figure 7. I suggest removing the first paragraph entirely, and just focusing on the behavior as a function of latitude and day of year, which clearly shows that the MHWs are more prevalent in the winter (northern and southern).

We agree and have removed that short paragraph.

Figure 10. It is impossible to distinguish between the four different colors of green, and the two different yellows. Is there even much difference in the distribution between the negative and positive phases? If there is, it is impossible to tell on this figure. I suggest redoing with just 5 colors, one per index.

We have changed the colors of this figure so as to better highlight the differences between climate phases. We kept the 10 indices (and added those suggested by the second reviewer), since the positive and negative phases of a climate mode are associated with a different distribution of compound events (Fig. 8 and 9 in the revised manuscript).

Be consistent in the use of acronyms. Acronyms are used for the climate modes in Figure 10, but in the text acronyms are not used. Same for LMF, acronym is defined but not always used.

We carefully checked the manuscript and now use acronyms consistently throughout the text.
* * *
**2 Response to Monique Messié**

**2.1 General comments:**

The paper by Le Grix et al. is a global analysis of marine heatwaves (MHW), low chlorophyll events (LChl), and most importantly, compound events defined as both occurring simultaneously. The authors characterize these events in terms of intensity, duration, and frequency, describe their spatial and temporal patterns (including seasonal cycle and interannual variability), and analyze their link with well-known climate indices. This is an excellent paper, well-written and easy to follow. The results are novel and this is a welcome study, particularly in a context where MHWs have been extensively studied but their association with reduced oceanic productivity less so. I do have 2 concerns detailed below that should be very easy to address but may significantly impact the results and some of the paper's conclusions.

The authors are certainly welcome to not apply these suggestions, in which case this choice needs to be carefully justified (including within the paper as I expect other readers

will have similar concerns).

We thank the reviewer for the positive and encouraging feedback. We have applied most of the suggestions, and if not (i.e., the duration threshold for the definition of extremes), we justified our choice in the main manuscript.

**2.2 Specific comments:**

One of the strengths of this paper is its continuity with the literature, particularly Holbrook et al. (2019) who analyzed MHWs. Intensity, duration and frequency are defined similarly, and the link with climate indices is conducted following the same method (contrasting event frequency in a positive or negative phase). Fig. 10 is even constructed similarly to Fig. 3b in Holbrook et al (2019), with most colors matching. This makes it easy to compare results from this paper with results from Holbrook et al., which is very good. There are 2 ways this continuity should be further improved in my opinion, for consistency but also because results may be significantly impacted.

First, I would recommend that a duration threshold be used, at least for MHW and LChl events. Holbrook et al. used 5 days, following recommendations by Hobday et al (2016). Currently, MHW, LChl, and compounds events can be as short as one day. This goes against previous recommendations by Hobday et al (2016) and their qualitative definition of a MHW as a "discrete *prolonged* anomalously warm water event". While no definition exists for LChl in the literature that I am aware of, it does make sense to use a similar definition. While including short-duration events can be justified (and using a given threshold does introduce a bias too), my concern is that the lack of threshold is also certainly the reason for the "heavily skewed" duration distribution mentioned l. 123 that required the use of the 90th percentile, rather than mean, for describing duration patterns. Based on the "heavily skewed" distribution for duration, I am concerned that most of the results (averaged frequency/intensity) are heavily skewed, too, towards short-duration events that can be considered to hardly qualify as MWH, LChl or compound events. Fig. 3d and Fig. 4b suggest that very few LChl and compound events > 5 days may be found over large parts of the ocean (where the 90th percentile is below 5 days); this is OK and an interesting result in itself. I understand that retaining all events justifies the 1% threshold for compound events; however, this threshold could be re-calculated at each pixel as the percentage of time the pixel is in a MHW multiplied by the percentage of time the pixel is in a LChl. The results could then be displayed as not only frequency (as in Fig. 4a), but also LMF relative to this local threshold as defined by Eqn (1). If you need to retain the < 5-day events, this needs to be very carefully justified – in particular, does it makes sense to consider that 10% of days at any location belong to a MHW event, and (another) 10% to a LChl event? Some figures should also be added to clarify how the results are impacted by these short-duration events. In particular, do frequency/intensity patterns change when only retaining events > 5 days?

Different definitions for MHWs have been used in the literature and many definitions (for example Frölicher et al. 2018; Laufkötter et al. 2020 as well as the IPCC SROCC report (Collins et al. 2019)) do not apply a duration threshold. Of course, if the reader

wants to compare our results with the Holbrook et al. 2019 study, having a similar definition would be advantageous. However, as we provide Figure A4 with the MHW and LChl events individually, there should be no need to compare with Holbrook et al. 2019. An additional reason, why we refrain from changing the definition, is the fact that there is no evidence that a 5-day threshold for LChl events and compound events is more impact relevant than any other threshold.

We added the following paragraph in the Methods section:

*"Here, we do not apply a duration threshold as has been done for example in Hobday et al. (2016) for MHWs. Duration thresholds are rather arbitrary as it is unknown which thresholds are most impact-relevant in particular for LChl events and compound events. Our definition without a duration threshold is consistent with the usage in the IPCC SROCC report (Collins et al. 2019)."* (l. 121-124)

In general, extreme event durations are typically exponentially distributed and therefore have a heavy tail. The few very long events strongly affect summary statistics such as the mean value. We thus consider presenting the 90th percentile a more robust and informative statistic to highlight spatial differences in the event duration. We added the following sentence in the Methods section where we define the duration metric:

*"Figure A3 in the Appendix shows that the duration of MHWs, LChl and compound events is exponentially distributed."* (l. 138-139)

Second, additional climate indices used by Holbrook et al. should be included, specifically the PDO and NPGO that both have strong footprints in the northern Pacific (Holbrook et al Fig. 3b). While these modes are decadal, they both display positive and negative phases during the 1998-2018 time period so there is no reason why they could not be analyzed. Considering how prevalent the PDO and NPGO are in the Pacific, including them makes sense and may change some of the Fig. 10 results (if Holbrook's results are any indication, I would expect the NPGO to replace the NAO in the north-eastern Pacific, and the PDO to replace the EMI in the north-western Pacific). The NPGO may not have enough negative values over 1998-2018 but the impact of the positive phase could be assessed at least – particularly if the positive/negative phases where compared to the neutral values, rather than the mean (which I believe makes more sense anyways as the mean can be skewed towards positive or negative events).

We agree and have included PDO and NPGO in our study. We added the following sentences to the manuscript:

*"the Pacific Decadal Oscillation (PDO), the North Pacific Gyre Oscillation (NPGO)"* (l. 153)

*"The positive phase of PDO is associated with increased frequency of compound events in the eastern equatorial Pacific, in the northeastern Pacific, and in the Indian ocean, and vice versa during the negative phase of PDO (Fig. 8e,f). Although the ENSO and*

*PDO patterns are very similar in the Pacific - PDO is often described as the long-lived El Niño-like climate pattern in the Pacific (Zhang et al., 1997) - they differ in the Southern Ocean where PDO phases are associated with less frequent compound events than ENSO phases (Fig. 8a,b,e,f). NPGO is another leading mode of climate variability in the Pacific; its positive phase is associated with suppressed compound event occurrence in the northern Pacific gyre and reduced occurrence in the southern Pacific gyre while its negative phase is associated with increased occurrence of compound events in the subtropical Pacific (Fig. 8g,h). Note that ENSO, EMI, PDO and NPGO are all correlated, their definition being based on climate variability in the Pacific Ocean."* (l. 280-288)

*"The positive phase of PDO is associated with the greatest occurrence of compound events in some parts of the Indian Ocean and of the tropical and northeastern Pacific. The negative phase of NPGO is associated with the largest frequency of compound events in some parts of the North Pacific gyre. NAO is associated with their highest occurrence in the eastern equatorial Atlantic, in the Gulf Stream region, and in a few parts of the northeastern Pacific."* (l. 301-305)

*"The PDO index is obtained by linearly regressing monthly SST anomalies upon the leading Principal Component of SST anomalies in the North Pacific Ocean poleward of 20°N (http://research.jisao.washington.edu/pdo/PDO.latest). The NPGO index is based on the second mode of sea surface height variability in the northeast Pacific and it accurately describes the climate pattern south of 40°N (www.o3d.org/npgo/npgo.php)."* (l. 417-420)

**2.3 Minor comments and technical corrections:**

l. 87-88: This is mentioned in the discussion, but it would be worth mentioning here that daily satellite chlorophyll cannot be used for this analysis because the data coverage is too poor at the daily scale (notably due to clouds).

We added to the mansucript:

*"For chlorophyll, satellite data derived from ocean colour cannot be used because the coverage is too poor at the daily scale, notably due to clouds."* (l. 88-89)

Fig. 2: Were time series smoothed using a 14-day running mean prior to MHW, LChl, and compound event definition? the text only mentions smoothing the daily seasonal cycle. If time series were not smoothed in the calculations, they should not be smoothed in the figure. If they were, please update the text.

We decided to smooth the time series on Fig. 2 to obtain a better visualization of how MHW, LChl, and compound events are defined, even though time series of SST and chl anomalies are not smoothed in the rest of our study. We added a figure with the original data (without smoothing) in the Appendix.

[Figure]

Figure 1: Without smoothing

[Figure]

Figure 2: With smoothing

We specified in the caption of Fig. 2 that the smoothing was done *"for illustrative purposes"*, and added the following sentence in the paragraph describing Fig. 2: *"Time series of SST and chlorophyll concentration are smoothed with a 14-day running mean to obtain a better visualization of extreme events; for comparison, figure A1 in the Appendix shows results with no smoothing."* (l. 116-118)

l. 123: see comments above regarding the heavily skewed distribution. If you decide to retain short-duration events, at the minimum the distribution should be displayed (eg. box plot) for readers to understand how much of an impact short-duration events may have on the results.

See our response higher up. We have added a figure showing the distribution of event durations in the Appendix. They are typically exponentially distributed with a heavy tail. As all our analysis consider days in an extreme/compound event separately, very long events have a similar effect on our results than many very short events. We thus consider the effect of including all short-term events minor. Note that we do not use the term "skewed" anymore, as it conveyed a somewhat misleading image of the duration distributions.

l. 159: as a suggestion and as stated above, consider comparing the frequency of extreme event days over each climate phase to their frequency over the neutral phase, rather than over the complete 1998-2018 period. How did you attribute events spanning several phases between positive/neutral/negative, particularly events that might be long enough to span both positive and negative phases?

We attribute each day in the time series to a positive, neutral or negative phase, and then compute the frequency of compound event days over the positive, neutral and negative phases. Therefore, we consider separately the days of a common event spanning multiple phases.

Please note that we now compare the frequency of extreme event days over the positive and negative phases of each climate mode to their frequency over the neutral phase. We added to the manuscript the following two sentences:

*"To estimate whether a climate mode has a discernible effect on local univariate and compound extreme events, we compare at each grid cell the frequency of extreme event days over the positive and negative phases to their frequency over the neutral phase."* (l. 160-162)

*"Figure 8 presents the frequency change in compound event days during these climate modes compared to their frequency over all days in a neutral phase from 1998 to 2018."* (l. 265-267)

l. 185 "Similarly to MHWs": I would actually argue, based on Fig. 3c vs d, that MHW and LChl have almost exactly opposite duration patterns over most of the global ocean when excluding the Southern Ocean.

We agree and have replaced *"Similarly to MHWs, short LChl events are located in the mid-latitudes"* by *"When excluding the Southern Ocean, MHW and LChl events have rather opposite duration patterns over most of the global ocean".* (l. 188-189)

l. 190 "MHWs and LChl events often occur simultaneously": In addition to comparing the compound event frequency to the expected frequency, it would be useful to report the percentage of MHWs events that coincide with a LChl event, and the percentage of LChl events that coincide with a MHW event ("day(s)" could be used instead of "event(s)" in this sentence, not sure which would be most informative).

This follows directly from the definition of compound events. At each grid cell, both MHWs and LChl events have a frequency of 10% of days. If compound events occur x% of days, the percentage of MHWs (LChl) that coincide with LChl (MHWs) events is $(x/10) * 100$ %. No changes are made in the manuscript.

Fig. 4: consider using a linear scale, or at least displaying "logical" colorbar ticks (e.g. 0, 1, 2, 3). As it is the results are difficult to visualize.

We now use a linear scale on Fig. 4.

l. 196-204: while the fact that compound events are located in regions where Chl/SST are negatively correlated makes perfect sense, this is still a very interesting result and it was nicely demonstrated.

Many thanks!

l. 210 "There are exceptions however. Some exceptions also occurred. . .": Should one of these 2 sentences be removed? Not sure what you meant here.

We wanted to say that the frequency pattern of compound MHW and LChl events shown in Fig. 4a is not coherent with Fig. 1 everywhere. Compound events are not frequent over all regions where prominent MHWs were associated with negative chlorophyll anomalies. We now write:

*"There are exceptions however, such as in the northern subtropical Pacific gyre where chlorophyll concentrations were locally high during the 2013-2015 MHW, even though compound MHW and LChl events are relatively frequent ($>$ 1.8% of all days) there."* (l. 213-215)

l. 217 "Long compound events ($>$ 10 days)": did you mean "where 10% of events last longer than 10 days"?

Indeed, we need to be more accurate here and have replaced *"$>$ 10 days"* by *"where 10% of events last longer than 10 days".*

l. 291-300: As acknowledged in the discussion, correlation does not equate causation. Please rephrase sentences such as "El Nino Modoki leads to the greatest occurrence. . .", "The Indian Ocean Dipole is the main contributing climate mode. . .", "The North Atlantic Oscillation is the main modulator. . .". The modes are associated with high compound frequency (as said in other sentences) but we don't know if they drive them.

We now write:

*"ENSO seems to be the main modulator of compound events in the eastern equatorial Pacific and in the northwestern part of the Indian Ocean, where El Niño events are associated with the highest frequency of compound event days from 1998 to 2018. The positive phase of PDO is associated with the greatest occurrence of compound events in some parts of the Indian Ocean and of the tropical and northeastern Pacific. The negative phase of NPGO is associated with the largest frequency of compound events in some parts of the North Pacific gyre. The Indian Ocean Dipole is the climate mode associated with the highest occurrence of compound events around Indonesia and in parts of the subtropical Pacific. NAO is associated with their highest occurrence in the eastern equatorial Atlantic, in the Gulf Stream region, and in some parts of the northeastern Pacific. Finally, AAO is associated with the highest frequency of compound events in some parts of the Southern Ocean."* (l. 299-306)

l. 313 and 320: aren't these sentences contradictory? the first indicate that the eastern equatorial Pacific is an exception to the Hayashida rule, the second indicate that the eastern equatorial Pacific behaves as expected. It almost seems like you consider the eastern equatorial Pacific to be nutrient-rich l. 313 and nutrient-limited l. 320.

The eastern equatorial Pacific is an exception in the sense that even though the background nutrient concentration is high, marine heatwaves are associated with low chlorophyll in this region. On l.320, we replaced:

*"we also identify elevated compound event frequency in the nutrient-limited surface waters of the low latitudes (including the eastern equatorial Pacific) and low compound event frequency in the nutrient-rich surface waters of the Southern Ocean (Fig. 4a)."*

by:

*"we also identify elevated compound event frequency in the nutrient-limited surface waters of the low latitudes and in the eastern equatorial Pacific, and low compound event frequency in the nutrient-rich surface waters of the Southern Ocean (Fig. 4a). The eastern equatorial Pacific behaves like a nutrient-limited region even though it is nutrient-rich."* (l. 332-335)

l. 321-323: any hypothesis as to what may be at play in these regions?

We added the following sentence:

*"There, phytoplankton growth may be limited by other key nutrients (e.g. iron around Antarctica) or increased phytoplankton grazing may lead to low chlorophyll during marine heatwaves."* (l. 337-338)

l. 337-338: this is not what I see when comparing Fig. 10 to Holbrook et al (2019, their Fig. 3b) for the Southern Ocean. Both figures highlight ENSO and the AAO/SAM, and both display a very complex picture. Did you refer to the patchiness and frequency of white pixels, indicating that no clear signal can be identified?

We agree that figures are not so different in the Southern Ocean and replaced *"over the Southern Ocean"* by *"in the western Pacific"* (l. 352), where ENSO is for example associated with the highest occurrence of compound events and EMI with the highest occurrence in MHWs.

l. 340: see comment #2. Particularly for the PDO, there are both positive and negative phases during 1998-2018.

We replaced:

*"This indicates again that other processes (see above) may affect chlorophyll concentrations in these regions or that MHWs are mostly modulated by climate modes that we omitted in our study (e.g. the Pacific Decadal Oscillation in the northeastern Pacific) because our shorter period of analysis does not capture their variability."*

by:

*"This indicates again that other processes (see above) may affect chlorophyll concentrations in these regions, that MHWs are mostly modulated by climate modes that we omitted in our study (e.g. the Interdecadal Pacific Oscillation in parts of the Pacific Ocean) because our shorter period of analysis does not capture their variability, or that some climate modes would be dominant if we used a longer period of analysis such as in Holbrook et al. 2019."* (l. 354-357)
* * *
**3 Response to the Associate Editor**

I have carefully read your letter which seems to address most of the comments and suggestions raised by both Referees. Yet, some of your responses are somewhat partial. This is the case in particular for the followings:

We thank the editor for her positive assessment and suggestions.

Question by Referee 1 « Figure 2. Why is just the time period 2013-2015 shown? And why this location?) » : Your response addresses this question but my understanding is that you do not intent to modify the manuscript accordingly, while it should be as

readers may have the same concern was the Referee.

We do not intend to show the full time period on Figure 2, because readers would not be able to read the figure anymore. Instead, we show a short time period at one grid cell to facilitate the visualization of univariate and compound extreme events on a single graph. Nevertheless, we added a figure with the original data (without smoothing) in the Appendix.

Question by Referee 2 «I would recommend that a duration threshold be used... » : In your response you indicate that « a 5-day threshold for LChl events and compound events is more impact relevant than any other threshold ». My impression is that, although acceptable, this response is quite vague and thus not compelling. If possible, I suggest you bring some quantitative elements to this response and/or relevant references.

This must be a misunderstanding. We wrote that there is NO evidence that a 5-day threshold for LChl events and compound events is more impact relevant than any other threshold. Even for marine heatwaves, a duration threshold is not always applied (see for example Frölicher et al. 2018 or Laufkötter et al. 2020). In fact, our definition is consistent with the marine heatwave definition used in the recent IPCC SROCC report, which does not include any duration threshold (Weyer 2019): "A marine heatwaves is a period of extreme warm near-sea surface temperature (SST) that persists for days to months and can extend up to thousands of kilometres." No changes are made to the manuscript.

Question by Referee 2 « Second, additional climate indices used by Holbrook et al. ... » : Your response is that you agree with this comment and included the proposed climate indices in your analysis. Yet, there is no mention of actual changes in the text or figures. Even, your response to the Referee's suggestion about l. 340 suggests you do not intent to bring modifications to your manuscript « ... climate modes that we omitted in our study (e.g. the Pacific Decadal Oscillation... ». Please clearly account for this important comment and address it carefully both in the revised manuscript and the responses-to-referees document.

Thank you very much for this pointer. We planned to modify our manuscript according to this review comment, but did not explicitly mention our planned modifications to the revised manuscript yet. We added to the manuscript (and to the response to the reviewer above):

*"the Pacific Decadal Oscillation (PDO), the North Pacific Gyre Oscillation (NPGO)"* (l. 153)

*"The positive phase of PDO is associated with increased frequency of compound events in the eastern equatorial Pacific, in the northeastern Pacific, and in the Indian ocean, and vice versa during the negative phase of PDO (Fig. 8e,f). Although the ENSO and PDO patterns are very similar in the Pacific - PDO is often described as the long-lived El Niño-like climate pattern in the Pacific (Zhang et al., 1997) - they differ in the Southern Ocean where PDO phases are associated with less frequent compound events than ENSO*

*phases (Fig. 8a,b,e,f). NPGO is another leading mode of climate variability in the Pacific; its positive phase is associated with suppressed compound event occurrence in the northern Pacific gyre and reduced occurrence in the southern Pacific gyre while its negative phase is associated with increased occurrence of compound events in the subtropical Pacific (Fig. 8g,h). Note that ENSO, EMI, PDO and NPGO are all correlated, their definition being based on climate variability in the Pacific Ocean."* (l. 280-288)

*"The positive phase of PDO is associated with the greatest occurrence of compound events in some parts of the Indian Ocean and of the tropical and northeastern Pacific. The negative phase of NPGO is associated with the largest frequency of compound events in some parts of the North Pacific gyre."* (l. 301-303)

*"The PDO index is obtained by linearly regressing monthly SST anomalies upon the leading Principal Component of SST anomalies in the North Pacific Ocean poleward of 20°N (http://research.jisao.washington.edu/pdo/PDO.latest). The NPGO index is based on the second mode of sea surface height variability in the northeast Pacific and it accurately describes the climate pattern south of 40°N (www.o3d.org/npgo/npgo.php)."* (l. 417-420)

Question by Referee 2 on l. 159 « How did you attribute events spanning several phases... » : Again, your response addresses the question but it does not seem you intend to modify the manuscript accordingly. I recommend you do so as readers may have the same question as the Referee.

We added the following sentences to the manuscript:
*"To estimate whether a climate mode has a discernible effect on local univariate and compound extreme events, we compare at each grid cell the frequency of extreme event days over the positive and negative phases to their frequency over the neutral phase."* (l. 160-162)

*"Figure 8 presents the frequency change in compound event days during these climate modes compared to their frequency over all days in a neutral phase from 1998 to 2018."* (l. 265-267)

Finally, I suggest you add to the maps a reference grid indicating main parallels and meridians, which I think would be very useful to the readers (some of them will necessarily be interested in specific areas which they will want to locate easily).

We thank you for this suggestion, but would prefer not to include a reference grid on our maps to keep them as clean, clear and large as possible. Even in the IPCC reports and other high level assessments,such as in the Second World Ocean Assessment, parallels and meridians are usually not shown.

Please upload your revised manuscript, including the text, figures and supplementary material. Make sure that all the modifications are clearly highlighted (e.g., in a different color).

We now highlighted the modified text of the manuscript in *italics.*

Please also upload your responses to the Referees, including the lines and/or pages where you made modifications in order to accommodate the Referees' comments.

Done.

**References**

M. Collins, M. Sutherland, L. Bouwer, S.-M. Cheong, T. Frölicher, H. J. D. Combes, M. K. Roxy, I. Losada, K. McInnes, B. Ratter, E. Rivera-Arriaga, R. Susanto, D. Swingedouw, and L. Tibig. Extremes, abrupt changes and managing risk. in: Ipcc special report on the ocean and cryosphere in a changing climate, 2019.

T. L. Frölicher, E. M. Fischer, and N. Gruber. Marine heatwaves under global warming. *Nature*, 560(7718):360–364, 2018. ISSN 1476-4687. doi: https://doi.org/10.1038/s41586-018-0383-9.

W. W. Gregg and C. S. Rousseaux. Decadal trends in global pelagic ocean chlorophyll: A new assessment integrating multiple satellites, in situ data, and models. *Journal of Geophysical Research: Oceans*, 119(9):5921–5933, 2014. doi: https://doi.org/10.1002/2014JC010158.

N. J. Holbrook, H. A. Scannell, A. S. Gupta, J. A. Benthuysen, M. Feng, E. C. J. Oliver, L. V. Alexander, M. T. Burrows, M. G. Donat, A. J. Hobday, P. J. Moore, S. E. Perkins-Kirkpatrick, D. A. Smale, S. C. S. Thomas, and W. Thomas. A global assessment of marine heatwaves and their drivers. *Nature Communications*, 10(1):2624, 2019. ISSN 2041-1723. doi: https://doi.org/10.1038/s41467-019-10206-z.

T. P. Hughes, K. D. Anderson, S. R. Connolly, S. F. Heron, J. T. Kerry, J. M. Lough, A. H. Baird, J. K. Baum, M. L. Berumen, T. C. Bridge, D. C. Claar, C. M. Eakin, J. P. Gilmour, N. A. J. Graham, H. Harrison, J.-P. A. Hobbs, A. S. Hoey, M. Hoogenboom, R. J. Lowe, M. T. McCulloch, J. M. Pandolfi, M. Pratchett, V. Schoepf, G. Torda, and S. K. Wilson. Spatial and temporal patterns of mass bleaching of corals in the anthropocene. *Science*, 359(6371):80–83, 2018. ISSN 0036-8075. doi: 10.1126/science.aan8048. URL https://science.sciencemag.org/content/359/6371/80.

C. Laufkötter, J. Zscheischler, and T. L. Frölicher. High-impact marine heatwaves attributable to human-induced global warming. *Science*, 369(6511):1621–1625, 2020. ISSN 0036-8075. doi: 10.1126/science.aba0690.

N. Weyer. IPCC, 2019: Annex I: Glossary [Weyer, N.M. (ed.)]. In: IPCC Special Report on the Ocean and Cryosphere in a Changing Climate [H.-O. Pörtner, D.C. Roberts, V. Masson-Delmotte, P. Zhai, M. Tignor, E. Poloczanska, K. Mintenbeck, A. Alegría, M. Nicolai, A. Okem, J. Petzold, B. Rama, N.M. Weyer (eds.)]. In Press, 2019.